# MULTILINGUAL VISUAL SPEECH RECOGNITION WITH A SINGLE MODEL USING VISUAL SPEECH UNIT

## ABSTRACT

This paper explores sentence-level *Multilingual Visual Speech Recognition* with a single model for the first time. As the massive multilingual modeling of visual data requires huge computational costs, we propose a novel strategy, processing with visual speech units. Motivated by the recent success of the audio speech unit, the proposed visual speech unit is obtained by discretizing the visual speech features extracted from the self-supervised visual speech model. To this end, we introduce multilingual AV-HuBERT (mAV-HuBERT) by training the model on 5,512 hours of multilingual audio-visual data. Through analysis, we verify that the visual speech units mainly contain viseme information while suppressing non-linguistic information. By using the visual speech units as the inputs of our system, we pretrain the model to predict corresponding text outputs on massive multilingual data constructed by merging several VSR databases. As both the inputs and outputs are discrete, we can greatly improve the training efficiency compared to the standard VSR training. Specifically, the input data size is reduced to 0.016% of the original video inputs. In order to complement the insufficient visual information in speech recognition, we apply curriculum learning where the inputs of the system begin with audio-visual speech units and gradually change to visual speech units. After pre-training, the model is finetuned on continuous features. We set new state-of-the-art multilingual VSR performances by achieving comparable performances to the previous language-specific VSR models, with a single trained model.

## 1 INTRODUCTION

These days, speech processing technologies have made great progress in diverse applications such as speech recognition (Guo et al., 2021; Shim et al., 2021; Prabhavalkar et al., 2023; Hong et al., 2023), speech synthesis (Wu et al., 2022; Choi et al., 2023; Zhang et al., 2023; Jiang et al., 2023; Maiti et al., 2023), and speech translation (Inaguma et al., 2019; Jia et al., 2022; Lee et al., 2022; Kim et al., 2023a). Now, it is easy to find a speech processing model that can proficiently handle approximately 100 languages (Adams et al., 2019; Radford et al., 2023). However, multilingualism has been mainly explored for audio-based speech processing systems (Toshniwal et al., 2018; Lux et al., 2022), while visual-based speech processing systems are still tied to developing monolingual systems (Ma et al., 2022a; Kim et al., 2023c; Yeo et al., 2023c). There are two reasons for the lagging development of visual-based speech processing systems: 1) The high dimensionality of visual data compared to audio puts a challenge in training a large-scale model with massive multilingual data. Compared to the same length of audio, visual data requires about six times larger bits (Kim et al., 2023b) in a standard visual speech recognition process (Ma et al., 2021b). Moreover, the requirement of encoding spatial information using two-dimensional convolutions also increases the computation costs of visual speech processing compared to its counterpart. 2) The low quantity of labeled data in visual speech processing systems presents a formidable obstacle to technology development. In contrast to the tremendous amount of publicly available audio-text data (Pratap et al., 2020), a very limited number of video-text data are available, especially for non-English (Kim et al., 2023c).

In this paper, we explore the multilingualism of visual speech processing, especially in speech recognition (Assael et al., 2016; Petridis & Pantic, 2016; Chung & Zisserman, 2017a; Ma et al., 2021a; 2022b). Hence, our objective is to devise a multilingual Visual Speech Recognition (VSR) method that can recognize different languages with a single trained model. In order to mitigate the challenges in visual speech processing, we propose a novel strategy, processing with visual speech units.

The audio speech unit (Lakhotia et al., 2021) is a discretized representation of an extracted speech feature from a self-supervised speech model (Baevski et al., 2020; Hsu et al., 2021). It contains phonemic content (Sicherman & Adi, 2023) while suppressing the other speech characteristics (*e.g.*, speaker information) and can be employed as pseudo text. As it is the discretized signal of the original signal, the data size can be significantly reduced (Chang et al., 2023b; Kim et al., 2023b). Motivated by this, we propose visual speech unit, the quantized representation of the visual speech feature. As a result, one video frame having 61,952 bits (*i.e.*, based on a grayscale image with 88 × 88 size) can be expressed with one visual speech unit which can be represented with just 10 bits. With the huge data size reduction, 0.016% compared to the original, we can boost the training more than 10 times faster than the standard VSR training. Through analysis, we validate that the visual speech unit contains viseme information, the visual counterpart of phoneme, while suppressing the non-linguistic characteristics. Hence, enabling visual speech modeling using the visual speech units.

Specifically, we introduce multilingual Audio-Visual Hidden Unit BERT (mAV-HuBERT) which is the self-supervised speech model for extracting the visual speech unit. We train the mAV-HuBERT on 5,512 hours of multilingual audio-visual data composed of nine languages. Therefore, we can correctly put the multilingual viseme information into our visual speech unit. Then, we pre-train an encoder-decoder model by setting the inputs with the visual speech units and the outputs with the corresponding text, forming a unit-to-unit translation framework (*i.e.*, translation between discrete tokens) (Kim et al., 2023a). Inspired by the recent successes of VSR that leverage audio modal information to complement limited visual data (Zhao et al., 2020; Ren et al., 2021; Kim et al., 2021; Shi et al., 2021; Haliassos et al., 2022), we propose to use curriculum learning with a gradual increase in task difficulty using audio modality. Concretely, our unit-to-unit pre-training is initiated with audio-visual inputs and then gradually changed to visual inputs. With this curriculum learning, the model can find the optimization points stable and achieve higher performance with the complemented information of multi-modality. To mitigate the comparatively small amount of public visual-text paired data, we utilize the recently proposed automatic labels of Ma et al. (2023) and Yeo et al. (2023c) where the text labels are obtained by their automatic labeling processes. Finally, the pre-trained model is finetuned with continuous input features to maximize the VSR performances.

The major contributions of this paper can be summarized as follows: 1) To the best of our knowledge, this is the first work exploring sentence-level multilingual VSR with a single trained model. 2) We propose to use a visual speech unit which is a quantized visual speech feature extracted from a self-supervised model, in modeling visual speech representations. 3) In order to correctly capture the visemes from multilingual visual speech, we introduce a multilingual Audio-Visual Hidden Unit BERT (mAV-HuBERT) trained on 5,512 hours of multilingual data. 4) By employing the visual speech unit, we can train the multilingual VSR model with discrete inputs and outputs (*i.e.*, text). With this, we can drastically reduce the computational costs and accelerate the pre-training time by about 10 times compared to the standard training. 5) Through analysis, we verify that the visual speech unit mainly holds the viseme information while suppressing non-linguistic variations, enabling VSR training with discrete inputs. We set new state-of-the-art VSR performances by achieving comparable performances with the multiple previous monolingual VSR methods.

## 2 RELATED WORK

**Visual Speech Recognition (VSR)** aims to predict the spoken words from silent lip movements video. Early works (Chung & Zisserman, 2017b; Stafylakis & Tzimiropoulos, 2017; Petridis et al., 2017; 2018b) focused on word-level VSR by using CNN (He et al., 2016) and the RNN (Chung et al., 2014; Hochreiter & Schmidhuber, 1997). Large-scale lip-reading sentence datasets (Chung et al., 2017; Afouras et al., 2018b) have boosted the development of sentence-level VSR. By employing Transformer (Vaswani et al., 2017) architecture, Afouras et al. (2018a) proposed a powerful sentence-level end-to-end VSR model. Moreover, the integration of the hybrid CTC/Attention objective (Watanabe et al., 2017; Petridis et al., 2018a) into VSR, greatly improved the recognition performances. Recent VSR technologies (Ma et al., 2021b; Prajwal et al., 2022; Chang et al., 2023a; Ma et al., 2023) also employed transformer-variant architectures and improved the VSR performances. For advanced training strategies, many researchers try to reduce the gap between visual and audio modalities. They (Zhao et al., 2020; Afouras et al., 2020; Ren et al., 2021; Ma et al., 2021a; Kim et al., 2021; 2022; Yeo et al., 2023b) studied how to effectively transfer audio knowledge into the VSR model by using knowledge distillation (Hinton et al., 2015) and memory network (Weston

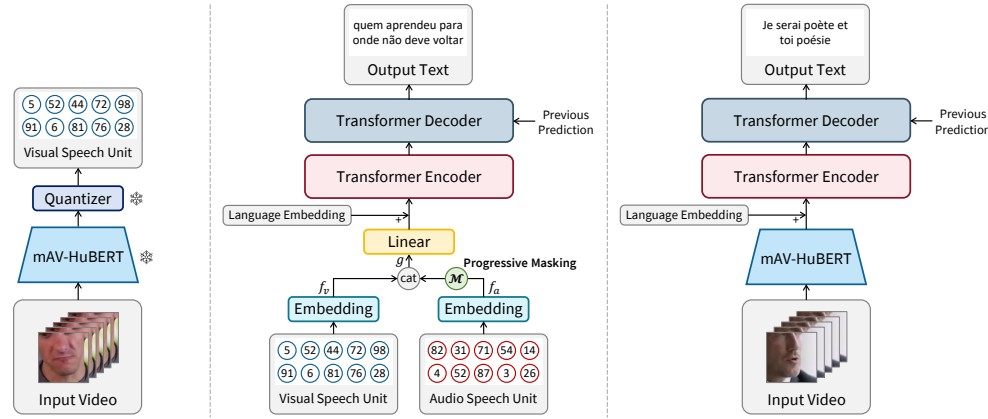

Figure 1: Illustration of the proposed multilingual VSR framework. (a) Visual speech units are obtained by quantizing the visual speech features extracted from mAV-HuBERT. (b) The Transformer encoder-decoder model is pre-trained with discrete inputs and outputs. The task difficulty is gradually increased by using progressive masking $\mathcal{M}$. Pre-training commences with audio-visual speech units as inputs, and these inputs gradually transition to visual speech units through progressive masking of the audio speech units. (c) After pre-training the model with discrete inputs and outputs, it is finetuned with continuous features to boost the VSR performances.

et al., 2015). However, these previous VSR approaches have mainly developed for high-resource languages, English and Mandarin (Luo et al., 2020). VSR for different languages, especially low VSR resource languages, has only been addressed recently (Ma et al., 2022a; Zinonos et al., 2023; Kim et al., 2023c; Yeo et al., 2023c). In particular, a recent approach (Yeo et al., 2023c) proposed the labeled data for low VSR resource languages using automatic labeling processes.

This paper is the first work exploring sentence-level multilingual VSR with a single model. To mitigate the huge computational costs in training the multilingual VSR model, we propose to pre-train the model with discrete inputs and outputs by using visual speech units. To complement the low amount of video-text data, we bring the automatic labels of Ma et al. (2023); Yeo et al. (2023c) and propose curriculum learning that utilizes audio modality to provide rich speech information.

**Audio speech unit** (Lakhotia et al., 2021) is the discretized speech representation of self-supervised speech models such as HuBERT (Hsu et al., 2021), Wav2Vec2.0 (Baevski et al., 2020), and WavLM (Chen et al., 2022). It is possible to suppress non-linguistic features and mainly keep the linguistic contents by selecting proper layers to extract the speech features (Lakhotia et al., 2021; Polyak et al., 2021). By using the speech unit as pseudo text, Textless Natural Language Processing becomes possible (Lee et al., 2022; Huang et al., 2022; Popuri et al., 2022; Kim et al., 2023a; Nguyen et al., 2023). Moreover, speech units have promising potential to be used in multi-modal processing as they greatly reduce the data size (Chang et al., 2023b; Park et al., 2023; Kim et al., 2023b).

Motivated by this, we propose a visual speech unit which is the quantized representation of visual speech features extracted from the self-supervised visual speech model (Shi et al., 2021). We introduce mAV-HuBERT for multilingual visual speech unit extraction. We show that the proposed visual speech unit contains mainly viseme information while suppressing the other characteristics.

## 3 METHOD

The objective of this paper is to develop a multilingual VSR model, so that multiple languages can be recognized by using a single trained model. To mitigate the large computational costs in developing visual speech processing systems, we propose visual speech units which are discretized representations of visual speech features encoded by a self-supervised speech model.

### 3.1 MULTILINGUAL VISUAL SPEECH UNIT EXTRACTION FROM SELF-SUPERVISED MODEL

Audio speech units (Lakhotia et al., 2021) can be obtained by clustering the speech features of a self-supervised speech model such as HuBERT (Hsu et al., 2021). Analogous to audio speech units,

we propose visual speech units, which can be obtained by quantizing the visual speech features derived from a pre-trained visual speech model. In order to get visual speech units, we choose AV-HuBERT (Shi et al., 2021) for the self-supervised visual speech model, which is well-known for its discriminative visual speech features. However, AV-HuBERT is pre-trained on English-only audio-visual data, which deviates from our primary objective of achieving multilingualism. Hence, we initially introduce a multilingual variant of AV-HuBERT (mAV-HuBERT) to ensure the accurate incorporation of multilingual viseme information into the final visual speech units. To this end, we train the model on 5,512 hours of multilingual dataset composed of 9 languages (En, Es, It, Fr, Pt, De, Ru, Ar, El) by merging LRS2 (Chung et al., 2017), LRS3 (Afouras et al., 2018b), VoxCeleb2 (Chung et al., 2018), and AVSpeech (Ephrat et al., 2018). As VoxCeleb2 and AVSpeech do not have language identities, we obtain the language identity of each utterance by using a pre-trained language identifier of Whisper (Radford et al., 2023), to select the data. For the prediction target of the masked prediction of mAV-HuBERT, we use clusters of speech features obtained from a pre-trained multilingual HuBERT (Hsu et al., 2021; Lee et al., 2022). We use the target size of 1,000 and train the model for 350k steps with one iteration. We show that our mAV-HuBERT is more suitable for multilingual speech modeling than the previous English-trained AV-HuBERT in Sec. 4.3.1.

With the pre-trained mAV-HuBERT, we extract the visual speech unit by clustering (*i.e.*, quantizing) the output visual speech features, as shown in Fig. 1(a). Please note that AV-HuBERT can extract both audio-only features and visual-only features through its modality dropout. Hence, we only use the visual inputs to extract the visual speech units. For the token size of the visual speech unit, we use 1,000 so that each visual speech unit can be represented with just 10 bits. Please note that one video frame having grayscale and $88 \times 88$ size (*i.e.*, the standard for visual speech recognition) requires 61,952 bits (Ma et al., 2021b; Shi et al., 2021). Therefore, we can reduce the data size to 0.016% compared to the raw visual inputs, which enables us to greatly increase the training batch size and reduce training speed by removing the visual front-end (*e.g.*, 2D CNNs). We analyze the efficiency of visual speech units by comparing it with the standard raw inputs in Sec. 4.3.2.

## 3.2 PRE-TRAINING: VISUAL SPEECH UNIT TO TEXT TRANSLATION

By setting the inputs with visual speech units, we pre-train our model to predict corresponding text. Therefore, now the inputs and outputs are both discrete, which is illustrated in Fig. 1(b). As the visual speech units mainly contain linguistic information, we can pre-train the model to construct the knowledge of visual speech modeling even by using discrete inputs. We analyze the information contained in the visual speech units and validate how it can be worked, in Sec. 4.3.3.

Nevertheless, translating visual speech units directly into output text from scratch is challenging for the model in identifying optimal solutions. Since visual information contains scarce speech information (Zhao et al., 2020; Kim et al., 2021; Ren et al., 2021) compared to audio, training the model directly to perform visual-to-text conversion might be hard to find the solution. To mitigate this, we bring the motivation from the recent success of VSR, which utilizes auxiliary audio information during training (Afouras et al., 2020; Zhao et al., 2020; Shi et al., 2021; Ren et al., 2021; Kim et al., 2022; Ma et al., 2022a; Yeo et al., 2023b). Specifically, we initiate the pre-training with audio-visual speech units where both audio speech units and visual speech units are utilized as inputs, similar to Shi et al. (2021); Djilali et al. (2023). Then, we gradually masked out the audio speech units as the pre-training progressed, resulting in the final training stage exclusively utilizing visual speech units as inputs. Therefore, the model can easily find the optimization points through this curriculum learning, with the aid of complementary audio speech information. Concretely, the embeddings of the visual speech units $f_v \in \mathbb{R}^{T \times D}$ and audio speech units $f_a \in \mathbb{R}^{T \times D}$ are concatenated as, $g = \mathcal{M}(f_a) \oplus f_v$, where $T$ is the sequence length, $D$ is the dimension of embedding, $\oplus$ represents concatenation operation in the embedding dimension, and $g \in \mathbb{R}^{T \times 2D}$ is the concatenated feature. $\mathcal{M}(\cdot)$ is a masking function that randomly masks out $p\%$ of frames from the input sequence. We progressively increase the $p$ from 0 to 100 as the pre-training progresses so that the transition from audio-visual inputs to visual inputs can be made. The effectiveness of this curriculum learning using input transition from audio-visual to visual can be found in Sec. 4.3.4.

Then, we reduce the embedding dimension of the concatenated feature $g$ using a linear layer. Here, we provide the language information by providing the language embedding which will be added to the feature, following Conneau & Lample (2019). Finally, through the Transformer encoder-decoder architecture, we translate the visual inputs into the output text in an auto-regressive manner.

The objective function of our learning problem can be represented as follows,

$$\mathcal{L} = -\sum_{s=1}^{S} \log P(y_s | \mathbf{X}, y_{<s}),$$ (1)

where $y_s$ is the text annotation for current step $s$ and $y_{<s}$ is the previous outputs, $\mathbf{X}$ is the input speech units, and $S$ is the length of the text. For multilingual training, we use five languages (En, Pt, Es, Fr, It) by merging LRS3 (Afouras et al., 2018b), mTEDx (Elizabeth et al., 2021), automatic labels for En of Ma et al. (2023), and automatic labels for Pt, Es, Fr, and It of Yeo et al. (2023c), forming 4,545 hours of data. We summarize the dataset statistics in Table 1.

### 3.3 Finetuning: Multilingual Visual Speech Recognition

Even though we can directly perform multilingual VSR with the pre-trained model with discrete inputs, it is hard to outperform the model using continuous features. This is expected, as information is lost during the quantization process. However, as the pre-trained model has already learned how to model the multilingual visual pronunciation and generate languages, the finetuning with the continuous features is straightforward and more effective than direct training of the multilingual VSR model from scratch. For finetuning, we detach the unit embedding and linear layers, and attach the pre-trained mAV-HuBERT, as illustrated in Fig. 1(c). The model is trained end-to-end with the same objective function and training data as the pre-training. In Sec. 4.3.5, we analyze the performances of multilingual VSR using the single proposed model and multiple previous monolingual models.

## 4 EXPERIMENT

### 4.1 DATASET

**Lip Reading Sentences 2 (LRS2)** (Chung et al., 2017) is one of the largest English datasets for VSR. The dataset consists of 223 hours of training data collected from British TV shows.

**Lip Reading Sentences 3 (LRS3)** (Afouras et al., 2018b) is a popular English VSR database. It has about 430 hours of video, and each video clip is collected from TED and TEDx. We evaluate the English VSR performances on LRS3.

**Multilingual TEDx (mTEDx)** (Elizabeth et al., 2021) is a multilingual dataset originally proposed for speech recognition and translation. The dataset provides 8 languages collected from TEDx talks. As the dataset also provides the video links of original talks, we download the video online. We remove the unavailable videos for VSR by referring to Ma et al. (2022a). We use four languages, Spanish (Es), Italian (It), French (Fr), and Portuguese (Pt) for training and evaluating the developed VSR model, following Ma et al. (2022a); Kim et al. (2023c); Yeo et al. (2023c).

**VoxCeleb2** (Chung et al., 2018) is a multilingual audio-visual dataset for speaker recognition (Jung et al., 2022). This dataset has over 1 million utterances and contains 6,112 celebrities. For mAV-HuBERT training, we use the data corresponding to the following 9 languages, English (En), Spanish (Es), Italian (It), French (Fr), Portuguese (Pt), German (De), Russian (Ru), Arabic (Ar), and Greek (El), by identifying the language identity using Whisper (Radford et al., 2023). For VSR training, as the dataset does not provide text annotations, we use the automatic labels of Ma et al. (2023) for En, and Yeo et al. (2023c) for Es, It, Fr, and Pt.

**Audio Visual Speech Dataset (AVSpeech)** (Ephrat et al., 2018) is a large-scale audio-visual speech dataset. AVSpeech contains roughly 290k YouTube videos, and the total duration of these videos is 4700 hours. Since the dataset does not provide text annotations, we use the same strategy with the VoxCeleb2, using 9 languages for mAV-HuBERT and automatic labels for VSR.

### 4.2 IMPLEMENTATION DETAILS

**Preprocessing.** The video is resampled to 25 fps. We detect the facial landmarks using RetinaFace (Deng et al., 2020), crop mouth regions using $96 \times 96$ sizes of bounding box, and convert them into grayscale. For data augmentation, we randomly crop the video into $88 \times 88$ and horizontally flip it during training. The audio is resampled to 16kHz. In order to obtain the audio speech unit, we

Table 1: Dataset statistics for pre-training mAV-HuBERT and training multilingual VSR model. Audio-visual data is utilized for training mAV-HuBERT, while video-text data is utilized for training the multilingual VSR model (*i.e.*, for both pre-training and finetuning).

| | Train Data for mAV-HuBERT | | | Train Data for multilingual VSR | |
| Datasets | Number of Video / Hours | Languages | Datasets | Number of Video / Hours | Languages |
|---|---|---|---|---|---|
| LRS2 | 142,157 / 223 | En | - | - | - |
| LRS3 | 150,498 / 433 | En | LRS3 | 150,498 / 433 | En |
| mTEDx | 181,034 / 285 | Es, Fr, It, Pt | mTEDx | 181,034 / 285 | Es, Fr, It, Pt |
| VoxCeleb2 | 834,375 / 1,739 | En, Es, It, Fr, Pt, De, Ru, Ar, El | VoxCeleb2 | 742,147 / 1,539 | En, Es, It, Fr, Pt |
| AVSpeech | 1,575,755 / 2,832 | En, Es, It, Fr, Pt, De, Ru, Ar, El | AVSpeech | 1,272,065 / 2,288 | En, Es, It, Fr, Pt |
| Total | 2,883,819 / 5,512 | En, Es, It, Fr, Pt, De, Ru, Ar, El | Total | 2,505,858 / 4,545 | En, Es, It, Fr, Pt |

Table 2: Comparisons between English AV-HuBERT and the proposed multilingual AV-HuBERT in multilingual modeling. English (En) is validated on LRS3 and other languages are validated on mTEDx databases.

| Method | Finetune Datasets | WER(%) | | | | |
| | | En | Es | It | Fr | Pt |
|---|---|---|---|---|---|---|
| AV-HuBERT (Shi et al., 2021) | LRS3, mTEDx | 28.0 | 75.9 | 74.0 | 75.5 | 79.6 |
| mAV-HuBERT (ours) | LRS3, mTEDx | 33.7 | 58.8 | 59.1 | 63.0 | 54.3 |

Table 3: Efficiency comparisons between previous VSR method and the proposed method. All numbers are measured using the same CPU and GPU (RTX 3090 24GB) environments. Test Acc refers to the subword-level prediction accuracy without beam search decoding.

| Method | Input | Batch Size (Tot. Frames) | Train Iter. Time (sec) | Tot. Train Time (hrs) | Test Acc (%) |
|---|---|---|---|---|---|
| Standard VSR | Video | 1,000 | 1.58 | 52.5 | 82.1 |
| Pre-training | Speech Unit | 6,000 | 0.88 | 6.6 | 72.2 |
| Finetuning | Video | 1,000 | 1.68 | 34.9 | 82.2 |

feed the audio into a multilingual trained HuBERT (Hsu et al., 2021; Lee et al., 2022) and cluster the extracted features with 1,000 token size. Finally, the audio speech unit is resampled to 25 fps to align with the sampling rate of the visual inputs. For the text, we construct one multilingual dictionary with 1,000 subword units by using SentencePiece tokenizer (Kudo & Richardson, 2018).

**Architecture.** Our mAV-HuBERT has the same architecture as the AV-HuBERT (Shi et al., 2021) large configuration. For visual speech unit-to-text translation, the model is composed of two unit embedding layers, one linear layer, one language embedding layer, 6 transformer encoders, and 6 transformer decoders. Each unit embedding layer embeds 1,000 tokens into 1,024 dimensions and the linear layer reduces the concatenated 2,048 dimensions into 1,024. The language token is embedded into a 1,024-dimensional feature by the language embedding layer. Each transformer layer has an embedding dimension of 1024, a feed-forward dimension of 4096, and 16 heads.

**Training.** For training mAV-HuBERT, we follow the original AV-HuBERT (Shi et al., 2021) and train it with the masked prediction task. For the prediction target, we use 1,000 clusters extracted from multilingual trained HuBERT (Hsu et al., 2021; Lee et al., 2022). We train the model for 350k steps using 64 3090 RTX GPUs. For pre-training the proposed model (*i.e.*, visual speech unit-to-text translation), we train the model for 11 epochs with a tri-stage learning rate scheduler and 32 GPUs. For the progressive masking $\mathcal{M}(\cdot)$, we set $p$ as 0 for the first 10% of training. From 10% to 70% of training, we linearly increase the masking ratio $p$ from 0 to 100. After 70% of training, $p$ is set to 100 so that only visual speech units are used. We finetune the model with the continuous features for 8 epochs using 32 GPUs. For all experiments, the Adam optimizer (Kingma & Ba, 2015) is used. For beam search decoding, we use a beam width chosen from {20, 25, 30, 35} and a length penalty of 0. The detailed training configuration can be found in the supplementary.

## 4.3 EXPERIMENTAL RESULTS

### 4.3.1 EFFECTIVENESS OF mAV-HuBERT IN MODELING MULTILINGUAL SPEECH

In order to verify the effectiveness of mAV-HuBERT in modeling multilingual speech, we compare our mAV-HuBERT with the English-trained AV-HuBERT. To this end, we finetune the pre-trained mAV-HuBERT and AV-HuBERT on 5 languages (*i.e.*, En, Es, It, Fr, Pt) using LRS3 and mTEDx databases. The multilingual VSR performances of the two models are shown in Table 2. The results show that when performing multilingual VSR with a single model, the mAV-HuBERT is much more effective than the English-only trained AV-HuBERT. Specifically, for Es, It, Fr, and Pt, the proposed mAV-HuBERT outperforms the AV-HuBERT with large margins of over 10% WERs. English VSR achieves better performance when utilizing English-trained AV-HuBERT. This is related to the curse

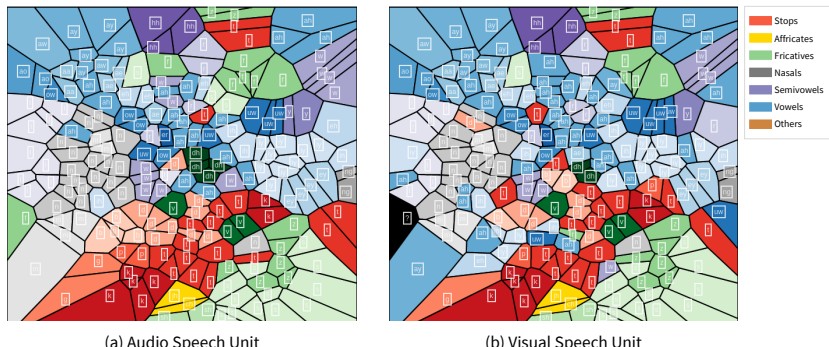

(a) Audio Speech Unit        (b) Visual Speech Unit

Figure 2: Visualization of speech units. Each boundary represents a single unit and the same color represents the same phoneme or phoneme family. (a) Audio speech unit. (b) Visual speech unit.

of multilinguality where an increasing number of languages can enhance cross-lingual performance up to a certain threshold, beyond which performance drops may occur (Conneau et al., 2020; Pfeiffer et al., 2022). As the proposed mAV-HuBERT shows better performances in modeling multilingual visual speech, we extract the visual speech unit by using the mAV-HuBERT.

### 4.3.2 EFFICIENCY COMPARISON BETWEEN VISUAL SPEECH UNITS AND RAW INPUTS

We compare the batch size, training iteration time, and total training time between the standard VSR method that uses raw video as inputs and the proposed method. Table 3 shows the comparison results. By using the visual speech units as inputs in pre-training, we can increase the batch size sixfold and reduce the training iteration time by approximately half. Consequently, we can expedite training by a factor of about 12 compared to previous VSR training methods. The total training time for pre-training the model for 11 epochs amounts to 6.6 hours, whereas the conventional VSR model requires 52.5 hours for 8 epochs of training. Furthermore, when fine-tuning the pre-trained model, we can achieve better performance compared to the standard VSR model, with just 5 epochs of fine-tuning taking 34.9 hours. As a result, we can greatly boost training time even if considering both the pre-training and finetuning stages, compared to the standard VSR training.

### 4.3.3 ANALYZING VISUAL SPEECH UNITS

To understand which information is held by the visual speech units, we analyze them using 1) phoneme mapping visualization (Sicherman & Adi, 2023) and 2) speaker recognition. Firstly, following Sicherman & Adi (2023), we visualize the phonetic information of the audio speech unit and visual speech unit which are obtained from mAV-HuBERT. Therefore, only audio inputs are used to extract audio speech units and video inputs for visual speech units. We set the cluster centroid as each row of the weight of the final layer of mAV-HuBERT (*i.e.*, classifier for 1,000 units), so that the same boundary can be obtained for different modalities. Fig. 2 displays the visualization results of the 200 units out of 1,000 that appeared most frequently. We can confirm that each unit in both the audio speech units and visual speech units contains distinctive linguistic information (*i.e.*, phoneme or viseme). By comparing the audio speech unit and visual speech unit, we can find that the homophenes which refer to the different pronunciations having the same lip movement (*e.g.*, name, tame, dame) are confusingly represented in visual speech units (Kim et al., 2022). For example, some units representing 'n' in audio speech units are changed to 'd' or 't' in visual speech units. Moreover, we can find that compared to audio speech units, more visual speech units are pointing to the vowel (*i.e.*, blue-colored area), which shows the ambiguity of lip movements compared to the audio modality. These are the natural results reflecting the characteristics of different speech modalities, and should not be taken as an indication of the inadequacy of the visual modality in speech modeling. With the visualization, we can confirm that our visual speech units contain the linguistic information, the viseme. More visualization results can be found in the supplementary.

Secondly, we analyze how much degree the speech units contain non-linguistic information through speaker recognition. To this end, we use raw audio speech, audio/visual speech features extracted from mAV-HuBERT, and audio/visual speech units as inputs to perform the speaker recognition. For the model, we follow Desplanques et al. (2020) and train the model on each input. The speaker

Table 4: Speaker verification results (EER) comparisons using different input representations.

| Raw audio | Audio Feature | Visual Feature | Audio speech unit | Visual speech unit |
|---|---|---|---|---|
| 2.38% | 14.96% | 19.42% | 28.84% | 32.74% |

Table 5: The effectiveness of each proposed component. By substituting each component from the full model, we evaluate their effectiveness in multilingual VSR.

| Method | En | Es | It | Fr | Pt |
|---|---|---|---|---|---|
| **Proposed Method** | 24.4 | 42.7 | 49.7 | 55.2 | 50.6 |
| − Unit Pretraining | 24.3 | 45.1 | 52.1 | 56.2 | 51.1 |
| − Curriculum | 25.2 | 46.3 | 52.5 | 55.8 | 50.6 |
| − Finetuning | 37.5 | 70.3 | 74.0 | 68.1 | 76.3 |

Table 6: Multilingual VSR performance comparisons. As there is no prior work that can perform multilingual VSR with a single model, we train AV-HuBERT to perform multilingual VSR.

| Method | En | Es | It | Fr | Pt |
|---|---|---|---|---|---|
| AV-HuBERT (Shi et al., 2021) | **23.3** | 51.2 | 54.8 | 61.0 | 55.3 |
| **Proposed Method** | 24.4 | **42.7** | **49.7** | **55.2** | **50.6** |

verification results (Equal Error Rate; EER) are shown in Table 4. The random prediction yields 50% EER and 0% EER means a perfect system. When we use raw audio as input, the system can almost perfectly distinguish the input speakers with 2.38% EER. When we use the features of the pre-trained mAV-HuBERT, the verification performance is dropped to 14.96% and 19.42% on audio and visual modalities, respectively. This shows that the masked prediction of AV-HuBERT (Shi et al., 2021) forces the model to learn linguistic information while somewhat discarding the speaker information. Finally, when we discretize the speech features of mAV-HuBERT and obtain the speech units, we can greatly suppress the speaker information. The performance is dropped to 28.84% by using the audio speech unit and 32.74% by using the visual speech unit. Through the above two experiments, we can confirm that the quantization process averages out the speaker effects (*i.e.*, non-linguistic information) in the speech representations while maintaining the content. This enables us to build a speech representation model by using speech units as input.

### 4.3.4 EFFECTIVENESS OF EACH PROPOSED COMPONENT

To confirm the effectiveness of each proposed component, we substitute each component from the proposed method. The ablation results are shown in Table 5. The performance of '−Unit Pretraining' is obtained by directly finetuning the proposed mAV-HuBERT on the multilingual video-text without using visual speech units. In this case, the overall VSR performances are dropped especially for non-English languages. Moreover, we require more training times compared to the proposed method as shown in Table 3. When we do not utilize curriculum learning (*i.e.*, '−Curriculum'), the performance dramatically decreases and even shows worse performances than the without pre-training method in some languages. This shows that directly performing the visual speech unit to text translation from scratch is challenging to the model in finding the optimal points. Therefore, the proposed curriculum learning is crucial when learning from the visual speech units. These results also coincide with the previous methods utilizing multi-modal complementary in VSR training (Afouras et al., 2020; Shi et al., 2021; Zhao et al., 2020; Ren et al., 2021; Kim et al., 2022; Yeo et al., 2023a). Finally, when we directly perform multilingual VSR with the pre-trained model (*i.e.*, with visual speech units), we cannot achieve satisfactory performances. Therefore, finetuning with the continuous features for a few epochs should be performed to maximize the pre-trained knowledge.

### 4.3.5 MULTILINGUAL VISUAL SPEECH RECOGNITION WITH A SINGLE TRAINED MODEL

We validate the effectiveness of the proposed multilingual VSR method by comparing it with 1) the multilingual VSR model and 2) the monolingual VSR model. Since there is no prior work exploring multilingual VSR with a single model, we train AV-HuBERT (Shi et al., 2021) to perform multilingual VSR and set it as the baseline. Moreover, since the previous non-English VSR methods are language-specific, we compare the performance of our model with multiple monolingual models.

**Comparison with multilingual VSR method.** Table 6 shows the performance comparison results of multilingual VSR methods. Both the AV-HuBERT and the proposed method are finetuned on 4,545 hours of multilingual video-text paired data. The proposed method outperforms the AV-HuBERT for all languages except English. In particular, the proposed method demonstrates significantly improved performance for Es, It, Fr, and Pt, with gains of more than 4% WERs. For the high-resource language En, the proposed method achieves similar performance with AV-HuBERT

Table 7: VSR performance comparisons with the previous state-of-the-art monolingual VSR methods. Please note that the proposed method utilizes one single multilingual model while the other methods utilize language-specific models. Best and second-best scores are bolded and underlined.

| Language | Method | Pre-training Data (hrs) | Language-specific Training Data (hrs) | Monolingual Model | Single Multilingual Model | WER(%) |
|---|---|---|---|---|---|---|
| En | Ma et al. (2022a) | - | 1,459 | ✓ | | 31.5 |
| | Prajwal et al. (2022) | - | 2,676 | ✓ | | 30.7 |
| | Shi et al. (2021) | 1,759 | 433 | ✓ | | 28.6 |
| | Zhu et al. (2023) | 1,759 | 433 | ✓ | | 28.4 |
| | Haliassos et al. (2022) | 1,759 | 433 | ✓ | | 27.8 |
| | Yeo et al. (2023a) | 1,759 | 433 | ✓ | | 27.6 |
| | Ma et al. (2023) | - | 3,448 | ✓ | | **20.5** |
| | **Proposed Method** | 5,512 | - | | ✓ | 24.4 |
| Es | Ma et al. (2022a) | 1,459 | 87 | ✓ | | 56.3 |
| | Kim et al. (2023c) | 3,448 | 72 | ✓ | | 56.9 |
| | Yeo et al. (2023c) | 3,448 | 384 | ✓ | | 45.7 |
| | **Proposed Method** | 5,512 | - | | ✓ | **42.7** |
| It | Ma et al. (2022a) | 1,459 | 46 | ✓ | | 57.4 |
| | Kim et al. (2023c) | 3,448 | 46 | ✓ | | 59.7 |
| | Yeo et al. (2023c) | 3,448 | 152 | ✓ | | 51.8 |
| | **Proposed Method** | 5,512 | - | | ✓ | **49.7** |
| Fr | Ma et al. (2022a) | 1,459 | 100 | ✓ | | 66.2 |
| | Kim et al. (2023c) | 3,448 | 85 | ✓ | | 64.9 |
| | Yeo et al. (2023c) | 3,448 | 331 | ✓ | | 58.3 |
| | **Proposed Method** | 5,512 | - | | ✓ | **55.2** |
| Pt | Ma et al. (2022a) | 1,459 | 99 | ✓ | | 61.5 |
| | Kim et al. (2023c) | 3,448 | 82 | ✓ | | 58.6 |
| | Yeo et al. (2023c) | 3,448 | 420 | ✓ | | **47.9** |
| | **Proposed Method** | 5,512 | - | | ✓ | 50.6 |

but slightly falls behind, as we have seen in Sec. 4.3.1. Considering multilingualism, the results confirm that the proposed strategy using visual speech units is much more effective in building multilingual VSR models by achieving new state-of-the-art multilingual VSR performances.

**Comparison with monolingual VSR method.** Here, we compare the proposed multilingual VSR performances with the previous state-of-the-art monolingual VSR methods. Please note that the proposed method utilizes a single trained model across the languages, while different methods utilize multiple language-specific VSR models. The results are shown in Table 7. By comparing the recent state-of-the-art method (Yeo et al., 2023c), we outperform it in 3 languages, except Portuguese (Pt). In the English VSR, our method achieves 24.4% WER while the current previous method (Ma et al., 2023) achieves 20.5% WER. Therefore, the proposed multilingual VSR model achieves the best score in Es, It, and Fr, and the second-best score in En and Pt with a single trained model. By analyzing the results, we can find that English (En) and Portuguese (Pt) have the largest portion of our training dataset. Similar to the results in Tables 2 and 6, we experience the curse of multilinguality (Conneau et al., 2020; Pfeiffer et al., 2022) that the performances for high-resource languages are dropped slightly while the performances for low-resource languages are greatly improved. Through the comparisons, we can confirm the effectiveness of the proposed method in multilingual VSR by outperforming and achieving comparable results with the previous language-specific VSR methods.

## 5 CONCLUSION

This paper proposed a multilingual VSR method using a single model. Specifically, we proposed to use visual speech unit modeling to mitigate the huge computational loads in building massive multilingual visual speech processing models. To this end, we introduced multilingual AV-HuBERT (mAV-HuBERT) trained on audio-visual data of nine languages. With the visual speech units, we can greatly reduce the data size and effectively pre-train the VSR model on large-scale multilingual VSR databases. By analyzing the visual speech unit, we validated it contains linguistic information and enables visual speech modeling using discrete inputs. To complement visual speech information with audio, we proposed curriculum learning by gradually increasing the task difficulty. By finetuning the model on continuous features, we set new state-of-the-art multilingual VSR performances by achieving comparable VSR performances with the previous language-specific VSR models.

## 6 ETHICS STATEMENT

We proposed a powerful VSR method that can recognize multilingual speech without accessing the audio. This technology can be applied for many beneficial applications such as recognizing speech in noisy environments or having conversations with patients who cannot make a voice due to surgeries. However, as all the technological advances risk being abused, the VSR technology can be misused in surveillance systems to recognize personal conversations from a distance by using a camera. Such misuse has the potential to infringe upon an individual's privacy. Therefore, it is imperative that the distribution and deployment of these technologies are carefully managed by both researchers and the research community.

## 7 REPRODUCIBILITY

In order to ensure reproducibility, we described in detail the training datasets, model architectures, proposed methods, and implementation details in Section 3, Section 4, and Supplementary. Moreover, the source code and pre-trained models for the proposed methods including mAV-HuBERT and multilingual VSR models will be publicly released to reproduce the results of the paper.

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

# SUPPLEMENTARY

## A VISUALIZATION OF VISUAL SPEECH UNITS

### A.1 VISUALIZATION OF SPEECH UNITS

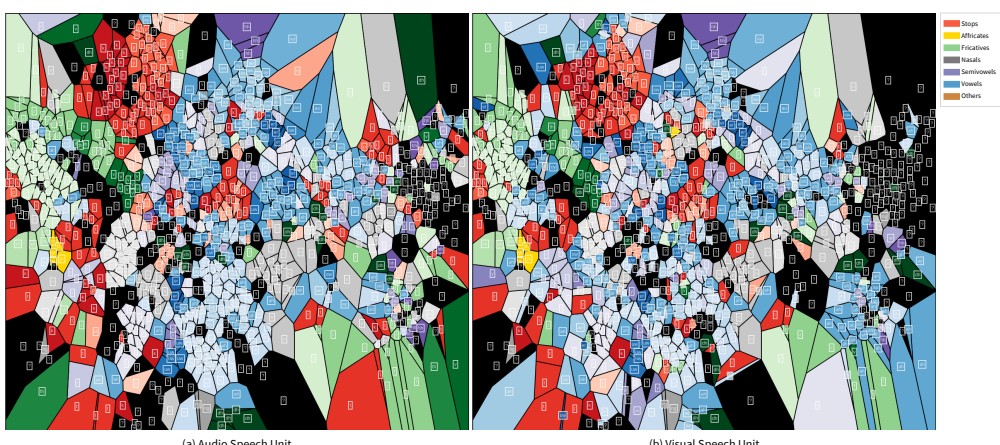

Figure 3: Visualization of speech units. Each boundary represents a single unit and the same color represents the same phoneme or phoneme family. (a) Audio speech unit. (b) Visual speech unit.

Fig. 3 shows the visualization of all 1,000 units of both the audio speech units and the visual speech units. In visual speech units, more units are classified as vowels while audio speech units have more distinct phonemes. As we discussed before, the ambiguity of lip movements is reflected in the figure.

### A.2 VISUALIZATION OF CORRESPONDING VIDEO FRAMES TO VISUAL SPEECH UNIT

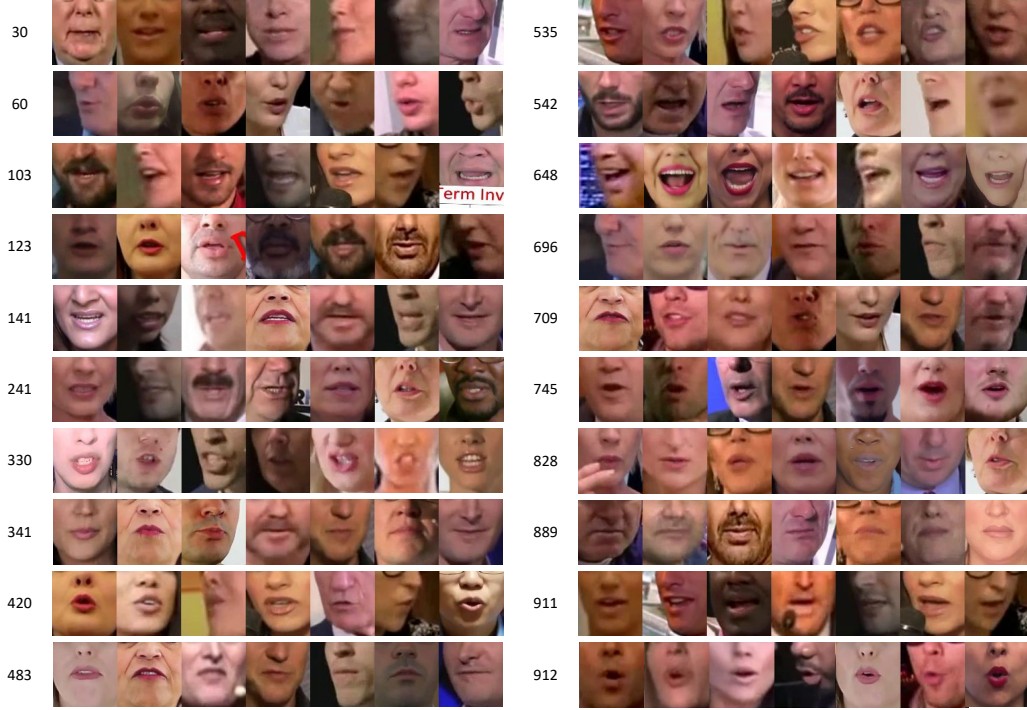

Figure 4: Visualization of video frames corresponding to visual speech units. Each number indicates an index of visual speech unit.

We visualize some video frames corresponding to each visual speech unit. The results of the visualization are depicted in Fig. 4, illustrating that similar lip movements are consistently mapped to the same index regardless of pose variations and speakers. For example, the 648-th visual speech unit represents lip frames related to the viseme 'a', and the 912-th visual speech unit corresponds to the viseme 'o'. Since visual speech units represent individual visemes, each conveying linguistic information, we can pre-train the model to build visual speech modeling knowledge.

# B  MONOLINGUAL VSR RESULTS

We also report the performances of monolingual finetuned models of the proposed method. Therefore, after pre-training using visual speech units, the model is finetuned on each language-specific training data. Furthermore, we also provide performance results achieved without employing pre-training, where the models are fine-tuned directly from our mAV-HuBERT. The results are presented in Table 8. We found that by finetuning the pre-trained model on each language, we can obtain better performance in English while the other languages show worse performance than using one multilingual model. Moreover, we can observe again that the performance improvements are marginal for high-resource languages but substantial for low-resource languages through multilingual training by achieving superior performances for non-English than the monolingual finetuned models. Finally, without the pre-training strategy using visual speech units as proposed, overall performance declines in both multilingual and monolingual finetuning results, confirming the effectiveness of the proposed method.

Table 8: Performances of the proposed method with language-specific finetuning and without pretraining.

| Language | Method | Pre-training Data (hrs) | Language-specific Training Data (hrs) | Multilingual Model | WER(%) |
|---|---|---|---|---|---|
| En | Proposed Method | 5,512 | 3,258 | ✗ | **23.7** |
| | − Unit Pretraining | 5,512 | 3,258 | ✗ | 24.3 |
| | Proposed Method | 5,512 | - | ✓ | 24.4 |
| | − Unit Pretraining | 5,512 | - | ✓ | 24.3 |
| Es | Proposed Method | 5,512 | 384 | ✗ | 44.9 |
| | − Unit Pretraining | 5,512 | 384 | ✗ | 48.5 |
| | Proposed Method | 5,512 | - | ✓ | **42.7** |
| | − Unit Pretraining | 5,512 | - | ✓ | 45.1 |
| It | Proposed Method | 5,512 | 152 | ✗ | 51.3 |
| | − Unit Pretraining | 5,512 | 152 | ✗ | 52.9 |
| | Proposed Method | 5,512 | - | ✓ | **49.7** |
| | − Unit Pretraining | 5,512 | - | ✓ | 52.1 |
| Fr | Proposed Method | 5,512 | 331 | ✗ | 56.6 |
| | − Unit Pretraining | 5,512 | 331 | ✗ | 57.6 |
| | Proposed Method | 5,512 | - | ✓ | **55.2** |
| | − Unit Pretraining | 5,512 | - | ✓ | 56.2 |
| Pt | Proposed Method | 5,512 | 420 | ✗ | 50.8 |
| | − Unit Pretraining | 5,512 | 420 | ✗ | 53.2 |
| | Proposed Method | 5,512 | - | ✓ | **50.6** |
| | − Unit Pretraining | 5,512 | - | ✓ | 51.1 |

# C  DETAILED EXPERIMENTAL SETUP

## C.1  DATASET STATISTICS FOR EACH LANGUAGE

The dataset statistics for each language are shown in Table 9. Please note that there are only 181,034 non-English human-labeled videos (*i.e.*, mTEDx). Therefore, we increase the quantity of labeled data by utilizing the automatic labels proposed by Yeo et al. (2023c); Ma et al. (2023). With this, we can construct 2,014,212 multilingual video-text paired data. The number of these automatic labels can be found in the 'Auto-labeled # of Video' column in the table.

Table 9: Statistics of datasets used in this work including automatic labels.

| Language | Dataset | Human-labeled # of Video | Auto-labeled # of Video | Hours | Total Hours |
|---|---|---|---|---|---|
| En | LRS2 | 142,157 | - | 223 | 3,481 |
| | LRS3 | 150,498 | - | 433 | |
| | VoxCeleb2 | - | 628,418 | 1,326 | |
| | AVSpeech | - | 837,044 | 1,499 | |
| Es | mTEDx | 44,532 | - | 72 | 384 |
| | VoxCeleb2 | - | 22,682 | 42 | |
| | AVSpeech | - | 151,173 | 270 | |
| It | mTEDx | 26,018 | - | 46 | 152 |
| | VoxCeleb2 | - | 19,261 | 38 | |
| | AVSpeech | - | 38,227 | 68 | |
| Fr | mTEDx | 58,426 | - | 85 | 331 |
| | VoxCeleb2 | - | 66,943 | 124 | |
| | AVSpeech | - | 69,020 | 122 | |
| Pt | mTEDx | 52,058 | - | 82 | 420 |
| | VoxCeleb2 | - | 4,843 | 9 | |
| | AVSpeech | - | 176,601 | 329 | |
| De | VoxCeleb2 | - | - | 190 | 333 |
| | AVSpeech | - | - | 143 | |
| Ru | VoxCeleb2 | - | - | 2 | 288 |
| | AVSpeech | - | - | 286 | |
| Ar | VoxCeleb2 | - | - | 7 | 114 |
| | AVSpeech | - | - | 107 | |
| El | VoxCeleb2 | - | - | 1 | 9 |
| | AVSpeech | - | - | 8 | |

## C.2 DETAILED TRAINING SETUP

We provide the detailed training setup used for experiments in Table 10. For pre-training mAV-HuBERT, we use a polynomial decay Learning Rate (LR) scheduler, batch size of 1,000 frames for each GPU, and train steps of 350k. For pre-training with the visual speech unit, we use 3,000 frames per GPU even though we can increase it to 6,000 frames. During finetuning, the pre-trained encoder is frozen for 10k steps and 7.2k steps for multilingual and monolingual finetuning, respectively.

Table 10: Details of hyperparameters used in training.

| | Pre-training (mAV-HuBERT) | Pre-training (Visual speech unit to text translation) | Fine-tuning (Multilingual) | Fine-tuning (Monolingual) |
|---|---|---|---|---|
| # of epochs | 40 | 11 | 8 | - |
| # of steps | 350,000 | 60,000 | 120,000 | 60,000 |
| # of frozen steps | - | - | 10,000 | 7,200 |
| # of GPUs | 64 | 32 | 32 | 8 |
| Max frames / batch | 1000 | 3000 | 1000 | 1000 |
| LR scheduler | polynomial decay | tri-stage | tri-stage | tri-stage |
| warmup updates | 48,000 | 15,000 | 15,000 | 15,000 |
| peak learning rate | 2e-3 | 1e-3 | 4e-4 | 4e-4 |
| Adam $(\beta_1, \beta_2)$ | (0.9, 0.98) | (0.9, 0.98) | (0.9, 0.98) | (0.9, 0.98) |

## C.3 DETAILED SPEAKER VERIFICATION SETUP

We utilize a pre-trained speaker recognition model of Desplanques et al. (2020) and train the model with different inputs. To match the input size with the model, we use one additional embedding layer for both audio speech units and visual speech units. When we use the speech features from mAV-HuBERT, we utilize one additional linear layer for both audio speech features and visual speech features. For training, we utilize data less than 20 seconds in VoxCeleb2 (Chung et al., 2018) dev set. The speaker verification is performed on the test data of VoxCeleb2 having less than 20 seconds, where each test sample is assigned one positive sample and one negative sample. Therefore, 27,816 positive pairs and negative pairs are utilized respectively, thus random prediction yields 50% EER.

## D  EFFICIENCY OF THE PROPOSED METHOD

We examine the subword-level accuracy changes during fine-tuning, comparing the proposed method with the model without pre-training (*i.e.*, '−Unit Pretraining'). Fig. 5 shows the learning curve of the two models. After the proposed pre-training using visual speech units, the finetuning on the continuous features is very effective so that we can achieve better performance even with much fewer epochs than the model without pre-training. It's worth noting that we can significantly reduce the pre-training time with the proposed method; 1 epoch requires just 0.6 hours in pre-training, whereas the standard VSR training demands 6.6 hours for 1 epoch.

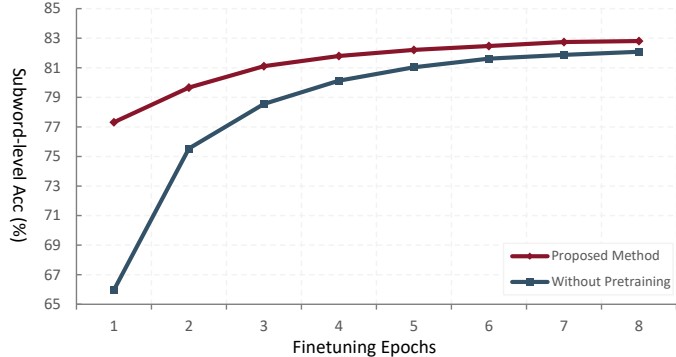

Figure 5: Finetuning efficiency comparison between the proposed pre-training scheme and without the pre-training.

## E  EXAMPLES OF PREDICTED SENTENCES

We show some examples of predicted transcriptions by the proposed multilingual VSR model and ground-truth transcriptions in Fig. 6. The model can perform multilingual VSR with a single model.

| | | |
|---|---|---|
| **English (En)** | Ground Truth: | the choices don't make sense because it's the wrong question |
| | Prediction: | choices don't make sense because it's the wrong question |
| | Ground Truth: | this is not a statement on malnutrition or anything else |
| | Prediction: | this is not a statement on malnutrition or anything |
| **Spanish (Es)** | Ground Truth: | si os digo la verdad hasta hace poco no me había hecho esa pregunta |
| | Prediction: | yo sigo la verdad que hasta hace pocos me había hecho esa pregunta |
| | Ground Truth: | los papelitos oficiales están a la venta desde ahora corran que se acaban |
| | Prediction: | los preparatos oficiales están en la venta desde ahora corren que sacaban |
| **Italian (It)** | Ground Truth: | ed è molto diversa dalle precedenti per almeno cinque motivi |
| | Prediction: | era molto diversa dalle precedenti perché erano cinque motivi |
| | Ground Truth: | ora andiamo nella parrocchia di quartiere a corso francia |
| | Prediction: | ora andiamo nella persona di quel diritto su francia |
| **French (Fr)** | Ground Truth: | et je vais vous en citer trois mais il y en a énormément |
| | Prediction: | et je vais vous enregistrer trois milieux énorméments |
| | Ground Truth: | ça cest le maîtremot de lévolution et des espèces qui veulent survivre |
| | Prediction: | ça cest de mettre le bonheur de lévolution des espèces qui veulent survivre |
| **Portuguese (Pt)** | Ground Truth: | e na puberdade a gente usa muito o método científico |
| | Prediction: | e nesse momento a gente usa muito mais um científico |
| | Ground Truth: | se um dia tu pudesse conhecer alguma pessoa quem seria essa pessoa |
| | Prediction: | se o jeito pudesse conhecer alguma pessoa crescer nessa pessoa |

Figure 6: Example sentences predicted from the single proposed multilingual VSR model on LRS3 and mTEDx test set. The Red and Blue indicate deletion and wrong predicted words, respectively.

