# OpenReview forum: "Multilingual Visual Speech Recognition with a Single Model using Visual Speech Unit"
_ICLR.cc/2024/Conference — Submitted to ICLR 2024_

### Official Review · Reviewer_RNNa · 2023-10-29

**Soundness:** 3 good
**Presentation:** 3 good
**Contribution:** 3 good
**Rating:** 6
**Confidence:** 3

**Summary:**

This paper proposed a visual speech recognition model that can perform recognition on multiple languages. The model consists of a AV-HuBERT module that extracts visual speech units, followed by transformer module that converts visual speech units into text. The proposed model is trained in three steps. The first step focus on training mAV-HuBERT module by regressing visual input to discrete visual speech unit with supervision coming from pre-trained multilingual HuBERT model. The second step conducts pre-training on transformer module by taking visual speech unit extracted by mAV-HuBERT model and predict text. The final step conducts an end-to-end fine-tuning of all modules by taking visual input and predict text. Experimental evaluation demonstrated that the proposed mAV-HuBERT model can effectively perform multilingual visual speech recognition compared to mono-lingual method which has to be trained on individual language. The proposed training strategy also improved over naive multilingual model trained on all languages while reducing training time by using discrete visual speech unit as input during pre-training.

**Strengths:**

1. This paper extended AV-HuBERT to effectively handle multilingual scenario. Compared to mono-lingual AV-HuBERT that is trained on individual language, mAV-HuBERT achieved best WER on three out of five languages and second-best on two out of five languages. The improvement on under-resourced languages is especially encouraging as the proposed method demonstrate a possibility to improve VSR on a specific language by leveraging other languages.
2. The proposed training strategy of mAV-HuBERT not only help improve the recognition accuracy of AV-HuBERT on different languages but also improves training efficiency by leveraging discrete visual speech unit during pre-training, which uses much less data storage and thus allows much higher batch size and reducing training time. The strategy of curating multilingual language dataset used during training is also a contribution
3. Analysis on visual speech unit provides insight on the unit such as the unit captures well on the viseme information rather than other information such as speaker identity.

**Weaknesses:**

1. Although the overall approach and the problem the paper tackles are novel, the core model is largely based on an existing model AV-HuBERT with minimal modification. The pre-training objective is also commonly used without much modification. Perhaps the authors could clarify a bit more on any contribution regarding extending AV-HuBERT in model architecture if applicable.
2. The comparison in Section 4.3.1 may not be completely fair. For AV-HuBERT, the model is pre-trained with English only, so fine-tuning on other languages means the English pre-trained AV-HuBERT is trained with same loss function on a new language, which will result in 5 different fine-tuned model i.e. one for each language. For mAV-HuBERT, a same model is supposed to work for different languages. So the fine-tuning is not supposed to be done on each language, which would yield 5 different models. If this is how the experiment was done, then mAV-HuBERT has advantage by design as it was pre-trained with more language data. If my understanding was not correct, then the authors should clarify on the specific process of fine-tuning of mAV-HuBERT.
3. The comparison with multilingual VSR approaches is weak. The only comparison done was with AV-HuBERT as the authors claim there is no prior work that can perform multilingual VSR with a single model. However, there are recent work and reference therein indicate exploration along this direction. For example,
- Cheng et al., MixSpeech: Cross-Modality Self-Learning with Audio-Visual Stream Mixup for Visual Speech Translation and Recognition, ICCV 2023
- Anwar et al., Muavic: A multilingual audio-visual corpus for robust speech recognition and robust speech-to-text
translation, 2023.

**Questions:**

1. I'm curious on how the number of visual token size used to determine visual speech unit used in the first step affect the final performance. Do the authors vary the number (1000 being used in the paper) and choose the one with better performance?
2. Regarding the curriculum learning, I'm also curious on how the learning schedule affect the performance. And when p% reaches 100%, the embedding from audio speech unit is useless. Do we disregard the embedding completely (thus no concatenation needed) or we still retain the same process to generate concatenated embedding?

---

> ### Author Response · Authors · 2023-11-19
> **Responses to Reviewer RNNa**
>
> ### Firstly, we would like to thank the reviewer for his/her valuable comment and time spent. We took all comments thoughtfully and will reflect them in our future modifications.
>
> > Below is the response to the comment.
>
> **(Q1) The contribution regarding AV-HuBERT**
>
> **(A1)** The mAV-HuBERT has the same model architecture as AV-HuBERT model. The main difference is the pre-training dataset. We trained the mAV-HuBERT on 5,512 hours of multilingual data (9 languages), while the original AV-HuBERT is trained on English-only data of 1,759 hours. Please note that by building mAV-HuBERT, we can greatly improve the VSR performances for multilingual speech while the original AV-HuBERT model shows insufficient ability for modeling languages other than English. We will release the pre-trained weight of mAV-HuBERT which we expect to boost the research stream in multilingual and multimodal speech processing.
>
> In addition to mAV-HuBERT, we would like to highlight the novelty of this paper.
>
> - To the best of our knowledge, **this is the first work exploring multilingual Visual Speech Recognition (VSR) using a single model**. Previous VSR models that explored multiple languages are language-specific, so the same number of models with the number of languages should be prepared to perform VSR for multiple languages. However, with the proposed method, we can perform it with just a single trained model. In this paper, we verified the effectiveness of the proposed method in 5 languages.
>
> - To the best of our knowledge, **this is the first work exploring visual-only speech units (i.e., visual speech units)**. Through analysis, we showed that visual speech units also correctly contain linguistic information similar to audio speech units. By employing the useful characteristics of the visual speech units (1. efficiency, 2. pseudo text), we first propose a method of pre-training the multilingual VSR model. We show the proposed method is very efficient and powerful through extensive evaluations.
>
> ---
>
> **(Q2) Clarification of Section 4.3.1**
>
> **(A2)** We would like to clarify how the AV-HuBERT was finetuned, in Section 4.3.1. In this section, our objective is to understand the multilingual speech modeling ability of AV-HuBERT and mAV-HuBERT. Therefore, we finetuned both AV-HuBERT and mAV-HuBERT with the same multilingual dataset with the same objective to perform multilingual VSR with a single model. Consequently, one AV-HuBERT model and one mAV-HuBERT model were utilized to measure the performances in Table 2 in the manuscript. Through the experiments, we showed that English-only trained AV-HuBERT is not appropriate to be directly utilized in multilingual modeling purposes and the proposed mAV-HuBERT is more appropriate for multilingual speech modeling.
>
> ---
>
> **(Q3) Prior multilingual VSR works**
>
> **(A3)** We would like to point out that the suggested references by the reviewer did not handle multilingual VSR.
>
> 1. The paper [1] first explored the visual-to-text translation task. Their model can be pre-trained with audio corpus by their proposed speech mixing. However, their lip reading setting is not multilingual. They trained each model on each language dataset (Table 3 of [1]).
>
> 2. The paper [2] mainly proposed a new multilingual audio-visual dataset. They only investigated the performances of Audio-based Speech Recognition (ASR) and Audio-Visual Speech Recognition (AVSR) on the proposed databases. Therefore, there was no multilingual VSR explored. Especially, they utilized the AV-HuBERT model directly and trained it on their databases to obtain ASR and AVSR. We would like to point out that no new architecture was proposed, and a dataset was proposed in their paper.
>
> Therefore, we would like to highlight that to the best of our knowledge, this is the first work exploring multilingual VSR using a single model. We will add the suggested references and discuss them in the manuscript.
>
> ---
>
> **(Q4) About the visual token size**
>
> **(A4)** We evaluated the purity of the clusters in advance to train our model, following [3]. The purity for different cluster sizes (200, 500, 1000, 2000) is shown in Table R5-1. Basically, because the clustering is the process of quantizing the continuous visual features into fixed cluster sizes, the more the cluster size yields better representation quality. Therefore, there is a trade-off between the performance and efficiency aspects. We used 1000 clusters which can be expressed with 10 bits, following the previous work [4].
>
> * **Table R5-1**. Clustering purity using different cluster sizes.
> | 200 | 500 | 1000 | 2000 |
> | :--------: | :--------: | :--------: | :--------: |
> | 20.71% | 31.06% | 35.38% | 43.51% |
>
> ---

---

> > ### Author Response · Authors · 2023-11-19
> > **Responses to Reviewer RNNa - 2**
> >
> > **(Q5) About the curriculum learning**
> >
> > **(A5)** We evaluated the pre-training performance by differing the $p$ before finetuning, and we observed that it is not critical in final performances, shown in Table R5-2. We choose the value showing the best mean performance. Please note that the important part is not the value of $p$, but whether we apply the curriculum learning or not, as shown in Table 5 in the manuscript. When we don't use the speech embedding, we replace the embedding part with zero vectors so the concatenation is still performed.
> >
> > * **Table R5-2**. Pre-training performances (WER %) using different $p$.
> > | **$p$** | **En** | **It**  | **Fr** |
> > | :--------: | :--------: | :--------: | :--------: |
> > | **50%** | 37.4 | 71.3 | 73.8 |
> > | **70%** | 37.5 | 70.3 | 74.0 |
> > | **90%** | 37.9 | 71.5 | 73.7 |
> >
> > ---
> > **References**
> >
> >     [1] Cheng et al., MixSpeech: Cross-Modality Self-Learning with Audio-Visual Stream Mixup for Visual Speech Translation and Recognition, ICCV 2023
> >     [2] Anwar et al., Muavic: A multilingual audio-visual corpus for robust speech recognition and robust speech-to-text translation, 2023.
> >     [3] Shi, Bowen, et al. "Learning Audio-Visual Speech Representation by Masked Multimodal Cluster Prediction." International Conference on Learning Representations. 2021.
> >     [4] Lee, Ann, et al. "Textless Speech-to-Speech Translation on Real Data." Proceedings of the 2022 Conference of the North American Chapter of the Association for Computational Linguistics: Human Language Technologies. 2022.

---

> ### Comment · Reviewer_RNNa · 2023-11-23
> **Response to rebuttal**
>
> Thanks the authors for responding to my review. Most of my concerns are addressed and the authors promised to incorporate the response into the final version. Considering the overall novelty and promising experimental results. I'm willing to change my rating from marginally below the acceptance threshold to be marginally above the acceptance threshold.

---

### Official Review · Reviewer_tK3Y · 2023-10-31

**Soundness:** 3 good
**Presentation:** 2 fair
**Contribution:** 2 fair
**Rating:** 6
**Confidence:** 4

**Summary:**

The paper extends a recent work on audio speech units to the audio-visual (AV) domain and tackle the associated challenge of the need for large amounts of labeled training and the computational cost. In doing to, they also extend the previous work on English language AV that leveraged visual phoneme units to a single-model solution for several popular languages.

**Strengths:**

1. There is solid engineering of the final system, with sufficient details to support reproducibility.
2. Builds on the recent state-of-the-art and leverages the nifty insights in training large scale ASR across languages and modalities.
3. Good amount of ablation studies to demonstrate the efficacy of various system components.
4. Source code, models, and samples (to be?) made publicly available.

**Weaknesses:**

1. The reference list and the related work sections are relatively misleading for the title of the paper. Either the term, "neural network/deep learning models," or a similar qualifier has to be added to the title and abstract or the related work should be expanded to include the historical work in the area, prior to the use of deep learning models.
2. Section 3, particularly 3.2 is hard to follow with it not being self-contained. I recommend bringing the results that justify the variety of modeling and training choices being described here into this section itself.
3.  This work leverages good tips and engineering practices in building the mAV-HuBERT solution but is obscures the focus from what is the big science/research delta between AV-HuBERT and mAV-HuBERT.

**Questions:**

1. What is the justification of hyperparameter choices such as, "We use the target size of 1,000 and train the model for 350k steps with one iteration?"
2. "...reduce training speed by removing the visual front-end."  Unsure why this is a good thing, unless you are referring to reducing the step size.
3. Why does the curse of multi-linguality only affecting the results for English? How does the relative sizes of training data in various languages affect this?
4. "After 70% of training, p is set to 100 so that only visual speech units are used." What is magical about 70?

**Details Of Ethics Concerns:**

No additional concerns with this work beyond what the ones that affect the publicly available data that this work leverages.

---

> ### Author Response · Authors · 2023-11-19
> **Responses to Reviewer tK3Y**
>
> ### Firstly, we would like to thank the reviewer for his/her valuable comment and time spent. We took all comments thoughtfully and will reflect them in our future modifications.
>
> > Below is the response to the comment.
>
> **(Q1) Related work before deep learning**
>
> **(A1)** We would like to thank the reviewer for the valuable comment. We will add the related work to include historical work in VSR, prior to the use of deep learning models. Moreover, we will clarify that the proposed method is focused on using deep learning models.
>
> ---
>
> **(Q2) Clarification of Section 3.2**
>
> **(A2)** As per the reviewer's comment, we will reorganize Section 3 so the manuscript can be more self-contained.
>
> ---
>
> **(Q3) Differences between AV-HuBET and mAV-HuBERT**
>
> **(A3)** The proposed mAV-HuBERT has the same architecture and objective function as AV-HuBERT. The main difference between the two models is whether they are trained using multilingual data or English-only data. mAV-HuBERT is trained on 5,512 hours of multilingual data composed of 9 languages while AV-HuBERT is trained on 1,759 hours of English-only data. We show that the original AV-HuBERT has insufficient ability to model multilingual speech as shown in Table 2. By pre-training the mAV-HuBERT with large-scale multilingual audio-visual data, we can greatly improve its performance in terms of multilingual VSR. Please note that we will release the pre-trained weight of mAV-HuBERT which can be utilized in diverse multilingual multimodal speech processing.
>
> ---
>
> **(Q4) The justification of unit size**
>
> **(A4)** We evaluated the purity of the clusters in advance to train our model, following [1]. The purity for different cluster sizes (200, 500, 1000, 2000) is shown in Table R4-1. Basically, because the clustering is the process of quantizing the continuous visual features into fixed cluster sizes, the more the cluster size yields better representation quality. Therefore, there is a trade-off between the performance and efficiency aspects. We used 1000 clusters which can be expressed with 10 bits, following the previous work [2].
>
> * **Table R4-1**. Clustering purity using different cluster sizes.
> | 200 | 500 | 1000 | 2000 |
> | :--------: | :--------: | :--------: | :--------: |
> | 20.71% | 31.06% | 35.38% | 43.51% |
>
> ---
>
> **(Q5) Clarification for the words "reduce training speed by removing the visual front-end"**
>
> **(A5)** We would like to clarify what the words "reduce training speed by removing the visual front-end" mean. Since we don't need the visual front-end (ResNet-18) for pre-training the visual speech unit-to-text model, the computation time can be reduced. This is because we don't need the gradient calculation using backpropagate the visual front-end. Therefore, the iteration time taken for each weight update step can be reduced.
>
> ---
>
> **(Q6) Why curse of multilinguality only affect English**
>
> **(A6)** The data proportion of English data for finetuning our model was about 70% (3,258hr/4,545hr), as shown in Table 1 and Table 9. As the English data is already large enough, the performance of the English-only trained model is very powerful. Therefore, it is not easy to outperform the English-only trained model with our multilingual-trained model which shares the model parameter for multi-task.
> If we finetune the mAV-HuBERT to English-only data, LRS3, making all model parameters devoted to English modeling, we can greatly improve the performance on English data as shown in Table R4-2.
>
> * **Table R4-2**. English performances of mAV-HuBERT finetuning on English-only and multilingual data.
> | Method | WER (%) |
> | :--------: | :--------: |
> | Multilingual Finetuned | 33.7% |
> | English-only Finetuned | 27.3% |
>
> ---
>
> **(Q7) About the number $p$**
>
> **(A7)** We evaluated the pre-training performance by differing the $p$ before finetuning, and we observed that it is not critical in final performances, shown in Table R4-3. We choose the value showing the best mean performance. Please note that the important part is not the value of $p$, but whether we apply the curriculum learning or not, as shown in Table 5 in the manuscript.
>
> * **Table R4-3**. Pre-training performances (WER %) using different $p$.
> | **$p$** | **En** | **It**  | **Fr** |
> | :--------: | :--------: | :--------: | :--------: |
> | **50%** | 37.4 | 71.3 | 73.8 |
> | **70%** | 37.5 | 70.3 | 74.0 |
> | **90%** | 37.9 | 71.5 | 73.7 |
>
> ---
> **References**
>
>     [1] Shi, Bowen, et al. "Learning Audio-Visual Speech Representation by Masked Multimodal Cluster Prediction." International Conference on Learning Representations. 2021.
>     [2] Lee, Ann, et al. "Textless Speech-to-Speech Translation on Real Data." Proceedings of the 2022 Conference of the North American Chapter of the Association for Computational Linguistics: Human Language Technologies. 2022.

---

> > ### Comment · Reviewer_tK3Y · 2023-11-22
> > **Acknowledging authors' responses**
> >
> > Thank you for the responses. Some of the changes should improve the clarity of the work.

---

### Official Review · Reviewer_SxNH · 2023-10-31

**Soundness:** 3 good
**Presentation:** 3 good
**Contribution:** 3 good
**Rating:** 8
**Confidence:** 4

**Summary:**

The paper proposes a multilingual visual speech recognition, by using the visual speech units to improve the training efficiency. It also proposes a curriculum learning approach to also exploits audio speech units. The paper presented number of experiments to investigate the effectiveness and efficiency of the proposed method.

**Strengths:**

- a novel frame work
- an efficient framework
- good presentation and experiments

**Weaknesses:**

- reproducibility seems difficult

**Questions:**

1- Computation aside, how different performance would be if we use continues features compared to unit features?

2- The paper mentions that one novelty is presenting the sentence level model. it would be good if some explanation is provided around advantage of sentence level models?

3- Once audio speech is utilised in the pertaining together with visual speech units, why using audio speech unit and not complete speech features? is it only to make computation more manageable? how performance and computation would change if audio speech is not discretised?

4- Table 2 contains interesting results, where the multilingual setup does not help English dataset. In addition to what authors mentioned, there could be other reasons like language similarities or available language specific information. This has been the topic of number of works around multilingual models that how to balance between more data Fram more languages or using less data only from fewer similar languages[1][2]. While I believe this paper's topic is not around this issue, I think it would be good to refer  to this point

[1] Cross-entropy training of DNN ensemble acoustic models for low-resource ASR
[2] Multilingual data selection for low resource speech recognition

**Details Of Ethics Concerns:**

no concern

---

> ### Author Response · Authors · 2023-11-19
> **Responses to Reviewer SxNH**
>
> ### Firstly, we would like to thank the reviewer for his/her valuable comment and time spent. We took all comments thoughtfully and will reflect them in our future modifications.
>
> > Below is the response to the comment.
>
> **(Q1) Computation aside, the performance if we use continuous features**
>
> **(A1)** Firstly, we would like to kindly remind the performances in Table 5 in the manuscript. If we don't use the continuous features at the inference and directly utilize speech unit features to inference, we cannot achieve enough performance because of the information lost during the quantization process ('-Finetuning' in Table 5). If we don't use speech unit features at all and train the model with continuous features in a standard VSR setting, we can get the performance of '-Unit Pretraining' in Table 5. By pre-training the model with unit features and finetuning the model on continuous features, we can get the best performance, as shown in Table 5 (Proposed Method). Please note that without the incorporation of visual speech units, the pre-training loses its distinctiveness and becomes the same with standard VSR training (continuous video features to text prediction).
>
> ---
>
> **(Q2) The advantage of sentence level models**
>
> **(A2)** We would like to thank the reviewer for the valuable suggestion. As lip movements (visual speech) have ambiguous speech information compared to audio and the existence of homophenes (same visual movements but different sounds; ex. ma, ba, pa), context modeling is crucial in the final performance. Therefore, using the sentence-level model brings the advantage of considering the context to predict the correct words and is thus more suitable in practical usage. In contrast, the famous word-level VSR model utilizes a very short video (1.16 sec in LRW [3]) wherein the target word must be centrally located and only can predict 500 words, making it challenging to apply effectively in real-world scenarios. As per the reviewer's suggestion, we will add the advantage of the sentence-level model in the manuscript.
>
> ---
>
> **(Q3) About using discrete audio speech units during pre-training**
>
> **(A3)** As the reviewer pointed out, the primary reason for using discrete audio speech units is computational efficiency. If we use raw audio inputs (log filterbank 26-dimension 100fps), the required bit size is 83,200bits for 1-second audio. If we use the speech units (cluster size 1000 25fps), the required bit size is 25bits for 1-second audio. Therefore, we can greatly reduce the computational and storage size by using the speech units. Moreover, we conclude that the performance gain will not be large by using continuous audio speech during the pre-training for the following reasons. 1) A recent study [4] showed that in audio-based speech recognition (ASR), discrete speech units can achieve similar performances with continuous features. 2) The objective of using audio inputs during pre-training is to help our visual-to-text learning, where the pre-trained model is finetuned on continuous visual features without audio. Hence, the more important thing is whether we used audio during pre-training as shown in Table 5.
>
> ---
>
> **(Q4) Other reasons why multilingual setup does not help English dataset**
>
> **(A4)** We thank the reviewer for providing valuable insights. We agree there can be other reasons why our multilingual setup does not help the English dataset than the curse of multilinguality.
>
> As demonstrated in the paper [5], one plausible explanation could be associated with model capacity. In our approach, we employed a model of the same size as the English-only model to handle multilingual speech. Scaling the model size could enhance the performance of high-resource languages as well.
>
> Also, another reason could be related to what the reviewer said, the similarity between languages. As the suggested references [1,2], using more similar languages will benefit the final speech recognition performances of each other, while not similar languages could deteriorate each other's performances.
>
> As per the reviewer's suggestion, we will add this discussion to the manuscript.
>
> ---
>
> **References**
>
>     [1] Sahraeian, Reza, and Dirk Van Compernolle. "Cross-entropy training of DNN ensemble acoustic models for low-resource ASR." IEEE/ACM Transactions on audio, speech, and language processing 26.11 (2018): 1991-2001.
>     [2] Thomas, Samuel, et al. "Multilingual Data Selection for Low Resource Speech Recognition." Interspeech. 2016.
>     [3] Chung, Joon Son, and Andrew Zisserman. "Lip reading in the wild." ACCV 2016.
>     [4] Chang, Xuankai, et al. "Exploration of Efficient End-to-End ASR using Discretized Input from Self-Supervised Learning." Interspeech. 2023.
>     [5] Radford, Alec, et al. "Robust speech recognition via large-scale weak supervision." International Conference on Machine Learning. PMLR, 2023.
>
> ---

---

> > ### Comment · Reviewer_SxNH · 2023-11-22
> >
> > thanks for the response

---

### Official Review · Reviewer_JHaL · 2023-11-05

**Soundness:** 3 good
**Presentation:** 3 good
**Contribution:** 2 fair
**Rating:** 5
**Confidence:** 4

**Summary:**

This work explores sentence-level Multilingual Visual Speech Recognition with a single model based on the pretrained AV-HuBERT and multi-lingual Hubert model. By taking advantages from the previous AV-HuBERT and the multi-lingual Hubert model together with the combination of multiple audio-visual speech datasets of different languages, the work here is able to transform the visual speech features to visual speech units and further perform multilingual visual speech recognition based on these units.

**Strengths:**

The general structure is clear. The method is simple in general. It’s easy to follow. It’s indeed very appealing in the view of efficiency when trained with discrete units.

**Weaknesses:**

Compared with new methods or other insights, I think this work is more like a case to show the success again of large-scale data and large-scale model. Similar to the audio speech units, the visual speech units are introduced here. But the concept of visual speech units has been introduced since AV-HuBERT and its contemporary works.
The method here is totally based on the pre-trained AV-HuBERT and multilingual HuBERT, with a common masking strategy by gradually increasing the masked ratio of audio samples. There seems no new points in the proposed method and training strategy. It is more like a presentation of an example to use existing large-scale models (AV-HuBERT, HuBERT).

**Questions:**

(1) In Table.2, the worse performance of the proposed method compared with AV-HuBERT is described as the curse of multilinguality. What’s the specific ratio of English data in the whole data? If the English data takes a large ratio in the whole data, and there are also common shared visemes among different languages, considering the larger-scale of the whole data compared with AV-HuBERT, the performance on English should not be worse?
(2) In Table 3, the proposed work is based on pre-trained AV-HuBERT and HuBERT. The “standard VSR” is also trained further based on these two models? or from scratch? The duration of “52.5 hours for 8 epochs” is also based on loading the pre-trained models?
(3) In Table 5, what’s the results of using only “unit pretraining”, whitout both CL and FT? Will it degenerate to the case of AV-HuBERT in Table 6?

---

> ### Author Response · Authors · 2023-11-19
> **Responses to Reviewer JHaL**
>
> ### Firstly, we would like to thank the reviewer for his/her valuable comment and time spent. We took all comments thoughtfully and will reflect them in our future modifications.
>
> > Below is the response to the comment.
>
> **(Q1) The novelty of the proposed method**
>
> **(A1)** We would like to highlight the novel parts of the paper.
>
> - To the best of our knowledge, there is no prior work explicitly utilizing visual-based discrete speech tokens and analyzed their characteristics. In this paper, we analyzed the characteristics of visual speech units and showed that they contain linguistic content while suppressing the speaker characteristics. By employing the characteristics, we proposed a novel pretraining strategy for multilingual VSR which is very efficient and powerful.
>
> - To the best of our knowledge, **this is the first work exploring multilingual Visual Speech Recognition (VSR) using a single model**. Previous VSR models that explored multiple languages are language-specific, so the same number of models with the number of languages should be prepared to perform VSR for multiple languages. However, with the proposed method, we can perform it with just a single trained model. In this paper, we verified the effectiveness of the proposed method in 5 languages.
>
> - Even though we utilized the same model architecture with AV-HuBERT, **we newly introduced multilingual AV-HuBERT (mAV-HuBERT)**. As we have shown the effectiveness of the mAV-HuBERT in multilingual speech modeling over English-only AV-HuBERT, we can contribute to the research society by providing the pre-trained mAV-HuBERT model. Please note that mAV-HuBERT is trained on about 5k hours of multilingual audio-visual datasets.
>
> ---
>
> **(Q2) The reason for worse English performance in Table 2**
>
> **(A2)** We would like to kindly remind the data size of each language is shown in Table 9 (Supplementary). The ratio of English data is  63% \(3,481hr/5512hr=0.63\). The curse of multilinguality is observed in diverse prior works exploring multilingual [1,2,3]. Especially, [1] shows that in order to outperform the English performance of the English-only trained speech recognition model with the multilingual trained speech recognition model, we need to scale the model size. [2] also finds the multilingual pretraining of speech recognition model improves the performance of non-English while decreasing the performance of English.
>
> These outcomes can be attributed to the substantial size of the English dataset in comparison to other languages. Consequently, achieving a performance that surpasses the English-only trained model proves challenging for the multilingual-trained model. In contrast, due to the small data size for low-resource languages, they can get benefits from other language datasets.
>
> In order to confirm the effectiveness of the mAV-HuBERT on English-only dataset, we finetune the mAV-HuBERT on English-only dataset (LRS3) and compare the performance with AV-HuBERT. The performance is shown in Table R2-1. The table clearly shows if mAV-HuBERT (trained on 5,512 hours multilingual data; English occupies 3,481 hours) is applied to an English-only corpus, we can obtain better performance than AV-HuBERT (trained on 1,759 hours of English data).
>
> * **Table R2-1**. VSR performance (WER %) comparisons on English-only dataset (LRS3)
> | Method | WER (%) |
> | :--------: | :--------: |
> | AV-HuBERT     | 28.6  |
> | **mAV-HuBERT**  | **27.3**  |
>
> ---
>
> **(Q3) About Standard VSR in Table 3**
>
> **(A3)** We would like to clarify how the Standard VSR in Table 3 is trained. Standard VSR refers to the model utilizing raw video inputs instead of the proposed visual speech units. Therefore, the Standard VSR is trained by initializing from our pre-trained mAV-HuBERT and then finetuned by attaching a decoder, without the proposed unit-to-text pre-training. In conclusion, the answer to the reviewer's question is "Yes, it also utilizes mAV-HuBERT which is pre-trained by setting its prediction target with mHuBERT, and the duration is also measured by using the pre-trained models". We will clarify it in our manuscript.
>
> ---
>
> **(Q4) The performance of only Unit Pretraining**
>
> **(A4)** As per the reviewer's comment, we report the performance of using only Unit Pretraining without both CL and FT. The performance is shown in Table R2-2.
>
> * **Table R2-2**. Performance (WER) without both CL and FT.
> | En | It | Fr |
> | :--------: | :--------: | :--------: |
> | 37.5 | 72.2 | 77.3 |
>
> Please note that if we don't perform finetuning on continuous visual inputs and directly utilize the unit-to-text pre-trained model for VSR, the input of the VSR system is now visual speech units, not raw video. Therefore, it does not degenerate to the case of AV-HuBERT in Table 6 which utilizes raw video inputs. Since the two systems' inputs are different (visual speech unit vs. raw video), AV-HuBERT performs better than the model "with only Unit Pretraining".
>
> ---

---

> > ### Author Response · Authors · 2023-11-19
> > **Responses to Reviewer JHaL (References)**
> >
> > **References**
> >
> >     [1] Radford, Alec, et al. "Robust speech recognition via large-scale weak supervision." International Conference on Machine Learning. PMLR, 2023.
> >     [2] Zinonos, Andreas, et al. "Learning Cross-Lingual Visual Speech Representations." ICASSP 2023-2023 IEEE International Conference on Acoustics, Speech and Signal Processing (ICASSP). IEEE, 2023.
> >     [3] Conneau, Alexis, et al. "Unsupervised Cross-lingual Representation Learning at Scale." Proceedings of the 58th Annual Meeting of the Association for Computational Linguistics. 2020.

---

### Official Review · Reviewer_uTmt · 2023-11-08

**Soundness:** 3 good
**Presentation:** 2 fair
**Contribution:** 2 fair
**Rating:** 6
**Confidence:** 3

**Summary:**

This paper proposes methods to train a multilingual visual speech recognition model, a first of its kind.  The main contributions of the paper are usage of quantized representation of AV-HuBERT embeddings to reduce the dimensionality of video data, alongside proposing a curriculum learning approach to improve the VSR performance. This approach gradually diminishes the reliance on audio data during training, ultimately leading to the model being trained solely on video embeddings.  The proposed methods significantly reduces training computational costs while enhancing Visual Speech Recognition (VSR) performance across multiple languages.

**Strengths:**

-> Trains a single model to lip read in multiple languages, which I believe hasn't been done before.

-> There is a clear improvement in the total training time required to train the mAV-Hubert due to the proposed quantization of AV-Hubert embeddings.

-> The proposed multi-lingual model maintains its effectiveness and stands its ground when compared to the monolingual VSR method.

-> The paper has good analyses on the information captured by the discretized visual units which is useful to the VSR community.

**Weaknesses:**

-> The novely of the article is limited. The authors use preexisting blocks such as AV-Hubert and previously proposed methods of curriculum learning to improve the VSR performance.

-> From Table 5, it looks like curriculum learning does not seem to have a significant impact on the VSR performance.

-> The paper contains grammatical errors and requires proofreading.

**Questions:**

In table 6, the authors showed that AV-HuBERT trained on multi-lingual data performs worse that mAV-Hubert for non-English dataset, thereby claiming that their proposed strategy of using visual speech units for training mAV-HuBERT is more effective in building VSR models. The claim would be better substantiated if the authors can apply the same strategy for training mAV-HuBERT on English only train dataset and show that it performs better than AV-HuBERT for English test set.

**Details Of Ethics Concerns:**

As the authors indicate in their paper, lip reading technology has potential to be mis-used in surveillance systems which could violate the privacy of the speaker.

---

> ### Author Response · Authors · 2023-11-19
> **Responses to Reviewer uTmt**
>
> ### Firstly, we would like to thank the reviewer for his/her valuable comment and time spent. We took all comments thoughtfully and will reflect them in our future modifications.
>
> > Below is the response to the comment.
>
> **(Q1) The novelty of the proposed method**
>
> **(A1)** We would like to highlight the novelty of our paper.
> - To the best of our knowledge, **this is the first work exploring multilingual Visual Speech Recognition (VSR) using a single model**. Previous VSR models that explored multiple languages are language-specific, so the same number of models with the number of languages should be prepared to perform VSR for multiple languages. However, with the proposed method, we can perform it with just a single trained model. In this paper, we verified the effectiveness of the proposed method in 5 languages.
>
> - Even though we utilized the same model architecture with AV-HuBERT, **we newly introduced multilingual AV-HuBERT (mAV-HuBERT)**. As we have shown the effectiveness of the mAV-HuBERT in multilingual speech modeling over English-only AV-HuBERT, we can contribute to the research society by providing the pre-trained mAV-HuBERT model. Please note that mAV-HuBERT is trained on about 5k hours of multilingual audio-visual datasets.
>
> - To the best of our knowledge, **this is the first work exploring visual-only speech units (i.e., visual speech units)**. Through analysis, we showed that visual speech units also correctly contain linguistic information similar to audio speech units. By employing the useful characteristics of the visual speech units (1. efficiency, 2. pseudo text), we first propose a method of pre-training the multilingual VSR model. We show the proposed method is very efficient and powerful through extensive evaluations.
>
> ---
>
> **(Q2) The effectiveness of the curriculum learning**
>
> **(A2)** We would like to point out the importance of curriculum learning. In Table 5, when we substitute the curriculum learning from our final model (i.e., -Curriculum) the performance is drastically decreased from the full model. The performances are even worse than those of without pretraining method (i.e., -Unit Pretraining). These results indicate that if we do not use the curriculum learning during our pretraining, we cannot get optimum performances. Therefore, the pretraining should proceed with the proposed curriculum learning to achieve the best performances for all languages. As we discussed in Section 4.3.4, the result coincides with the previous VSR methods highlighting the audio-visual complemental effects in VSR.
>
> ---
>
> **(Q3) Gramatical errors**
>
> **(A3)** As per the reviewer's comment, we will proofread the paper and fix the grammatical errors.
>
> ---
>
> **(Q4) The effectiveness of the mAV-HuBERT on English only train dataset**
>
> **(A4)** We greatly appreciate for the valuable suggestion. As per the reviewer's suggestion, we compare the VSR performances of AV-HuBERT and mAV-HuBERT on an English-only dataset, LRS3. Therefore, the mAV-HuBERT model is finetuned on LRS3 dataset with the same setting as AV-HuBERT. The performance comparisons can be found in Table R1-1. As the mAV-HuBERT is trained with about 5,500 hours of data (English occupies 3,481 hours), we can achieve better VSR performance than AV-HuBERT (trained on 1,759 hours of English) when it is only finetuned on the English-only dataset.
>
> * **Table R1-1**. VSR performance (WER %) comparisons on English-only dataset (LRS3)
> | Method | WER (%) |
> | :--------: | :--------: |
> | AV-HuBERT     | 28.6  |
> | **mAV-HuBERT**  | **27.3**  |
>
> ---

---

> ### Comment · Reviewer_uTmt · 2023-11-22
>
> I thank the authors for their detailed response.
>
> ---
>
> ### (Q2) The effectiveness of the curriculum learning
> Thanks for the clarification. I seemed to have read Table 5 wrong and I stand corrected. Curriculum learning does show a large impact.
>
> ---
>
> ### (Q4) The effectiveness of the mAV-HuBERT on English only train dataset
> The objective of the experiment with English only dataset was to understand the impact of the technique proposed by the author, namely "use a visual speech unit which is a quantized visual speech feature extracted from a self-supervised model, in modeling visual speech representations.". The experiment the authors conducted above does not address this issue. It looks like mAV-HUBERT has been trained on larger amount of data, which makes it harder to compare with respect to a strong baseline such as AV-HUBERT.
>
> I would propose the following:
>
> - Start with the same checkpoint for both mAV-HUBERT and AV-HUBERT
> - Use the same amount of English only data to fine tune mAV-HUBERT using the proposed quantized visual speech feature + finetuning as well as to finetune AV-HUBERT model
> - Compare the performance on English only dataset

---

> > ### Author Response · Authors · 2023-11-23
> > **Responses to Reviewer uTmt - 2**
> >
> > **(Q4) The effectiveness of the proposed "visual speech unit pre-training"**
> >
> > **(A4)** First of all, as the authors have limited time to run experiments suggested by the reviewer, it's hard to show the performance of the finetuned AV-HuBERT model by the authors. Instead, the authors can show the **effectiveness of the proposed pre-training method using the "visual speech unit"**, which is the reviewer's main concern. If we don't use the proposed pre-training method that uses visual speech units, and directly finetune the mAV-HuBERT on multilingual VSR data (which is the same as what AV-HuBERT proposed, except multilingual), the performances for Spanish, Italian, French, and Portuguese are decreased as shown in Table 5 ('- Unit Pretraining'). By applying the proposed pre-training method, we can improve the performances by about 3% WERs for Spanish and Italian, and 1% WERs for French and Portuguese. It is shown in Table 5 ('Proposed Method'). The experimental results clearly show the effectiveness of the proposed pre-training method using visual speech units.
> >
> > Moreover, we report the English-only finetuned result (LRS3) of the 'mAV-HuBERT + proposed visual speech unit pre-training'.
> > The obtained English-only finetuned performance is 25.4% WER. We can compare the performance with AV-HuBERT (28.6% WER) which is reported in the original AV-HuBERT paper. Therefore, we can clearly observe the benefits of the proposed visual speech unit pre-training in English-only data by outperforming the original AV-HuBERT model.

---

### Comment · Area_Chair_TkqT · 2023-11-10
**reviewer-author discussions**

Dear All,

The reviewer-author discussion period will be from Nov. 10 to Nov. 22. For reviewers, please read the authors' responses and acknowledge it, respond to them early on in the discussion, and discuss points of disagreement. Thank you!

AC

---

> ### Comment · Area_Chair_TkqT · 2023-11-21
> **Please discuss authors' response**
>
> Dear Reviewers,
>
> The authors have submitted the response to all reviews. Would you please read the authors' responses and acknowledge it, respond to them early on in the discussion, and discuss points of disagreement? Thank you!
>
> AC

---

### Meta-Review · Area_Chair_TkqT · 2023-12-05

**Metareview:**

The paper proposes a novel strategy for multilingual visual speech recognition (VSR) using visual speech units, which are discretized visual features obtained from a self-supervised model. The paper introduces a new model called multilingual AV-HuBERT (mAV-HuBERT), which is trained on a large-scale multilingual audio-visual dataset. The paper also applies curriculum learning to gradually shift from audio-visual inputs to visual-only inputs. The experiments clearly show the benefits of using multilingual data for representation learning. The main contributions of this paper are: 1) visual speech units, 2) curriculum learning strategy for this task, 3) application to multilingual scenarios with a single model. These contributions make the paper strong.

The common concern from reviewers is that the fundamental model is still AV-HuBERT, which is trained with multilingual data and becomes mAV-HuBERT. As the authors mentioned the proposed mAV-HuBERT has the same architecture and objective function as AV-HuBERT. This is of limited novelty, and the paper is like an application (multilingual) paper with engineering (curriculum learning) efforts. We don’t see too much algorithm contribution except the visual speech units. Although the authors emphasized the work is the first work exploring multilingual Visual Speech Recognition (VSR) using a single model, this sounds like a natural extension from the monolingual to the multilingual setup. The extension and the experimental discovery have already been shown in the speech recognition field, and the conclusion is expected.

Most reviewers also raised concerns on the results in the English set in Table 2. In the rebuttal Table R1-1, the authors provided another comparison which shows the fine-tuned mAV-HuBERT has better results than AV-HuBERT. However, this seems to be unfair comparison given mAV-HuBERT used much large amount of English training data.

**Justification For Why Not Higher Score:**

Although the paper has several strengths as noted above, however, the novelty is limited as the fundamental model is still AV-HuBERT, which is a prior art. Furthermore, there are some experimental results are questionable.

**Justification For Why Not Lower Score:**

N/A

---

### Decision · Program_Chairs · 2024-01-16

Reject